# *β*-Secretase-1: In Silico Drug Reposition for Alzheimer’s Disease

**DOI:** 10.3390/ijms24098164

**Published:** 2023-05-03

**Authors:** Roberto A. Galeana-Ascencio, Liliana Mendieta, Daniel I. Limon, Dino Gnecco, Joel L. Terán, María L. Orea, Alan Carrasco-Carballo

**Affiliations:** 1Laboratorio de Elucidación y Síntesis en Química Orgánica, ICUAP, BUAP, Puebla 72570, Mexico; roberto.galeanaa@alumno.buap.mx; 2Laboratorio de Neuroquímica, FCQ, BUAP, Puebla 72570, Mexico; liliana.martinezmen@correo.buap.mx; 3Laboratorio de Neurofarmacología, FCQ, BUAP, Puebla 72570, Mexico; 4Centro de Química, ICUAP, BUAP, Puebla 72570, Mexico

**Keywords:** *β*-secretase 1, in silico drug reposition, Alzheimer’s disease, molecular docking, ADME

## Abstract

The *β*-secretase-1 enzyme (BACE-1) performs a key role in the production of beta-Amyloid protein (Aβ), which is associated with the development of Alzheimer’s disease (AD). The inhibition of BACE-1 has been an important pharmacological strategy in the treatment of this neurodegenerative disease. This study aims to identify new potential candidates for the treatment of Alzheimer’s with the help of in silico studies, such as molecular docking and ADME prediction, from a broad list of candidates provided by the DrugBank database. From this analysis, 1145 drugs capable of interacting with the enzyme with a higher coupling energy than Verubecestat were obtained, subsequently only 83 presented higher coupling energy than EJ7. Applying the oral route of administration as inclusion criteria, only 41 candidates met this requirement; however, 6 of them are associated with diagnostic tests and not treatment, so 33 candidates were obtained. Finally, five candidates were identified as possible BACE-1 inhibitors drugs: Fluphenazine, Naratriptan, Bazedoxifene, Frovatriptan, and Raloxifene. These candidates exhibit pharmacophore-specific features, including the indole or thioindole group, and interactions with key amino acids in BACE-1. Overall, this study provides insights into the potential use of in silico methods for drug repurposing and identification of new candidates for the treatment of Alzheimer’s disease, especially those targeting BACE-1.

## 1. Introduction

Neurodegenerative diseases are a health problem in the medical sector that has increased considerably in recent years [1,2,3]. For Alzheimer’s disease, it is estimated that by 2050, there will be twice as many patients [4,5,6,7]. These diseases are associated with various molecular targets, one of them is an acetylcholinesterase (AChE), which has various drugs such as galantamine and donepezil [8,9,10,11,12]. Another target is the NMDA receptor, which is associated with memantine as a drug (Figure 1) [13,14,15]. Unfortunately, none of the above mentioned are cures for this disease, since they only stop the progression and are not completely efficient [3,16,17,18]. Therefore, the formation of the *β*-amyloid peptide, in which the *β*-secretase protease, particularly type 1, has been associated with an increase in the concentration of *β*-amyloid insoluble, which has a commercial inhibitor, verubecestat [19,20,21,22].

*β*-secretase-1 (BACE-1) is a type of membrane-bound aspartic protease, which is 501 amino acids in length (five key domains, a signal peptide, and pro-catalytic, catalytic, transmembrane, and cytoplasmic domains have been identified in the enzyme) [23]. Its crystalline structure reveals that the proteolytic pocket is relatively large, being able to accommodate up to 11 amino acid residues [23,24]. Amyloid plaques, composed mainly of *β*-amyloid peptide, are progressively formed in the brain of Alzheimer’s patients, and mutations in three genes cause the early onset of Familial Alzheimer’s disease (FAD) by directly increasing the toxic peptide that promotes *β*-amyloid 42 [25,26,27]. *β*-Amyloid is the product of the catabolism of a large protein, the *β*-Amyloid Precursor Protein (APP) [27,28,29]. Two proteases, *β* and *γ* secretase, endoproteolize the amyloid precursor protein (APP) to release the *β*-amyloid peptide [30,31,32]. In the brain of an Alzheimer’s patient, abnormal levels of APP accumulate, forming plaques that coalesce between neurons, disrupting cell function and promoting neuronal death [8,32].

Verubecestat and EJ7, however, are still experimental drugs, and not feasible options for clinical application [19,33,34]. A new approach that has taken great advantage in the case of SARS-CoV-2019 is drug repositioning [35], where drug databases are used to locate second options for medical application with minimal or no side effects [36,37]. A database with wide application in the United States and Latin America is Drugbank [38] with which, through in silico studies, the repositioning of the drug is proposed in this work, which is associated with BACE-1 inhibitors.

## 2. Results and Discussion 

It has recognized multiple experimental inhibitors, such as LY2811376, LY2886721, B67UFT75QS, Atabecestat, Lanabecestat, Elenbecestat, Umibecestat, Verubecestat, and EJ7, which present great particular coupling energies (See Appendix A), of which the extremes of activity are Verubecestat and the EJ7; these are bound to the catalytic site of *β*-secretase-1, explaining the inhibitory effect and proving the site of action (Figure 2). Thus, the interaction with the key amino acid residues, Asp31, Try70, Asp227, and Gly229 through hydrogen bonding, by the nitrogen in both structures, while the aromatic rings interact with the hydrophobic pocket around the residue Ile109 on the opposite side, both molecules under physiological conditions are protonated on the nitrogen of the heterocycle, which allows the formation of the hydrogen bond with Try70, a change that decreases BCE in its basic form. The higher energy observed by EJ7 is due to the hydrogen bonds formed by the first ring, since it does not present the electro-attracting group, EJ7 increases the strength of these intermolecular interactions; and the polar interaction that exists between the oxygen of the methoxide group with Asn232 instead of Fluor in the Verubecestat. In addition to the key interaction amino acids, when analyzing the structure of the reference drugs, an amide group is denoted in both structures (such as the peptide bond), which suggests the loss of effectiveness in these references since the structure can be fragmented, although this same group is the one that confers a great selectivity to the mimetic to the natural biological function of BACE-1.

Given the high structural similarity between Verubecestat and EJ7, it was possible to define the amino acid residues in the protein, and the energy limit for the search for BACE-1 inhibitors, as an alternative treatment for diseases, such as Parkinson’s, Alzheimer’s, and other neurological disorders. The Drugbank database has a large number of drugs, resulting from the modeling of each of these and the study of the coupling at the specific site of *β*-secretase-1, a total of 1145 with energy of higher coupling than Verubecestat (BCE = −4.30 kcal/mol), of which only 83 present better coupling energy than EJ7 (BCE = −6.05 kcal/mol), the latter being of greater interest given that when repositioning drugs, the clear objective is to improve existing options. The increase in the coupling energy is directly related to the increase in the inhibitory potential and, therefore, a decrease in the dose required to obtain the inhibitory effect of BACE-1, an increase of 1.0 kcal/mol is associated with a decrease in the dose by 50%, which makes these candidates better than Verubecestat and even better than EJ7. It is important to know the nature of these drugs, their current clinical use, routes of administration, and contraindications, before proposing them as a possible drug for pathologies associated with BACE-1 activity. Figure 3 shows the distribution of these drugs according to the anatomical-therapeutic-chemical classification code (ATC classification) [39] and the routes of administration available for them. Regarding the ATC classification, it allows us to analyze the type of prior use of the drugs candidates for repositioning, of the 83, 21% are not classified, given that their use is variable or is an adjunct in the diagnosis, which does not make them good candidates since its use is sporadic and on many occasions for single use. Group N, associated with the nervous system, with 9% of candidates, becomes of particular interest since they already have an association with pathologies in the nervous system; however, it must be analyzed that the adverse effects are not greater than the original condition or the reference drugs. The rest of the groups can be analyzed differentially by comparing their security windows and their capacity for prolonged use. Since the main use of BACE-1 inhibitors is mainly associated with the treatment for Alzheimer’s disease, a drug for this pathology must be easy to administer due to the need for continuous use, to improve cognitive functions and decrease the neurodegenerative process of patients. Therefore, the oral route of administration becomes a priority. Crossing the ATC classification and the route of administration, and the possibility of chronic use, allows to propose candidates that have a good route of administration obtaining 41 candidates that meet this requirement; however, 6 of these are associated with diagnostic tests rather than treatment, resulting in a total of 33 candidates that are described in Table 1.

When analyzing the pathologies, for which the 33 candidates are used (Table 1), those associated with infections, cancer, and the central nervous system (CNS) stand out. A priori, one might think that the latter is suitable for repositioning since it is known that they do affect the CNS, but the effect they have must be assessed; however, as, in the case of morphine, this drug tends to generate addiction, so its chronic use is contraindicated and the rest of opioids, the same case for ademetionine used for the treatment of chronic pain. Those associated with an antidepressant, aggressiveness, and schizophrenia treatment such as periciazine, mirtazapine, and perphenazine are not recommended for continuous use as required by Alzheimer’s. Another group that is not suitable for chronic use and for the elderly, who present the highest frequency of Alzheimer’s, are those associated with cardiac treatments since they can generate a lack of control of these functions in a patient without cardiovascular conditions, eliminating minoxidil, sotalol, and nebivolol. Finally, the third group that is not recommended is associated with diabetes, since most of these acts by decreasing high glucose levels, generating hypoglycemia in a patient without some type of diabetes and in a patient with this condition a competitive effect. Between the antidiabetic and the BACE-1 inhibitor, lypressin, canaglifloxin, and acarbose are not suitable for repositioning.

After the analysis of clinical use and route of administration, a total of 22 drugs are candidates for repositioning. The next criterion is to analyze their ADME properties (Table 2) to reposition them in ideal conditions for, after a preclinical study, being able to proceed to a Clinical Phase 2, since Phase 1 is already known. The next exclusion criterion is the use of Lipinsky’s rules, which predict oral bioavailability, for the route of administration and distribution, eliminating those that present two or more violations, such as desmopressin with 3three cefpiramide with two, and others. It results in a total of 13 candidates. Of which, 4 present a single rule violation, naldemedine by molecular weight, bazedoxifene, hydroxystilbamidine, and benserazide by the LogP or H-Bond value; however, for naldemedine and bazedoxifene the violation is minimal since they present a value close to the ideal, so they could be considered for second priority repositioning.

Finally, of the 9 primary candidates and the 2 secondary ones, it is necessary that these have good oral absorption and the ability to cross the blood–brain barrier (BBB), of which values of −3.0 to 1.2 are recommended and that they have a positive value in the prediction of activity in CNS, resulting in fluphenazine with a value of 2, naratriptan and bazedoxifene with a value of 1, frovatriptan and raloxifene with a value of 0. The 5 final candidates present a few metabolites from 3 to 7 and Jm values (excretion transdermal route) with frovatriptan being the best and bazedoxifene the worst, while the other 3 are in the same order; even so, they are acceptable values, guaranteeing low bioaccumulation, necessary for a drug that is proposed for chronic use. In addition, another advantage of these candidates is that they have active transport by P-glycoprotein, and the ability to BBB, which indicates a possible greater bioavailability and effectiveness on the nervous system. In addition, when analyzing the values of Jm and the polar topological area compared to the value of verubecestat, all the candidates are from the same area, except fluphenazine; however, in the metabolomic and excretion analysis, it is reported that these have less degradation; therefore, a longer half-life is expected, being another advantage of these candidates.

The 5 candidates energetically, due to their current clinical use and ADME properties, fulfill 4 at the primary level and 1 (bazedoxifene) secondary; however, it is necessary that said coupling be carried out at the catalytic site of *β*-secretase-1 and interact with amino acids, the key to the proteolytic process carried out on site. Figure 4A shows the active site of the enzyme and how it is occupied by the candidates, like the reference inhibitors. Overlapping of the aromatic rings is observed in the center of the catalytic site to proceed to generate the interactions around it, indicating the need for this anchorage in the inhibitor candidate.

When analyzing the interactions in the 2D diagrams (Figure 4B–F) it can be observed that the candidates present a directed amphipathic character, in the center by the aromatic part of the lipophilic properties and on the outside the result of the various substituents a polar character. At the level of hydrogen bonds, these are mostly mediated by the candidate as a donor towards Gln72, Lys106, Asp31, and Asp227, repeatedly and specifically from Try197 in fluvatriptan, Gly229 in bazedoxifene, Gly10 in frovatriptan and Gly 33 for saloxifene. Recapitulating, both verubecestat, and EJ7 interact by hydrogen bonding with Gly229, Asp31, and Asp 227. Given the presence of nitrogen atoms, under physiological conditions, these can be protonated, so the salt and pication bridges are of particular interest, in the reference inhibitors, Asp31 and Try70, respectively, the latter is in the form of Pi-Pi stacking form in fruvatriptan and raloxifene, while the other three candidates form salt bridges with Asp227. At the polarity level, there are other outstanding interactions in each of the candidates (Table 3), summarizing in all cases the amphipathic characteristic of the structures and an indole, thioindole, or tetrahydroquinoline nucleus, with the integration of a ring being a possible pharmacophore, aromatic bonded to a cyclopentene or cyclohexene with endocyclic nitrogen or sulfur. In addition, when analyzing the nature of the structures, it is highlighted that there is no formation of secondary amides, such as the peptide bond; thus, these candidates would not have a peptidomimetic similarity, which is indicative of a stronger inhibition, which hinders their metabolism and, therefore, a prolonged effect would be expected without a decrease in effectiveness due to degradation in the same enzyme.

Finally, to select an ideal candidate for repositioning, the final filter is their reported adverse effects, these should be at the same level or less than the drugs available or under study against BACE-1, particularly in this study. Table 4 is described the adverse effects of each of the candidates and of Verubecestat, the only reference inhibitor that already has these studies, denoting that the adverse effects range from mild to moderate, highlighting drowsiness, nausea, and numbness as common. While that the rest are special cases, with fluphenazine, raloxifene, and bazedoxifene being those whose side effects can be considered mild; in contrast, verubecestat’s most serious effects are suicidal ideation and anxiety, which in an elderly patient are of main concern, given the lifestyle they lead. In addition to minor effects, such as weight loss and dizziness, which normally result in falls and injuries, which although minimal in a young patient, in a hierarchical patient, are important given the low healing factors due to age. These candidates for repositioning, in addition to having lesser adverse effects than the reference ones, have phase 1 clinical studies and the safety window is known, although the next phase is to verify by in vitro studies, the most potent inhibitory effect by them, passing this filter and demonstrating the reduction in aggregates in amyloid plaques. These BACE-1 inhibitors will be available for new clinical studies and would have the advantage that safe doses are known for the design of new treatments and, thus, reduce the number of study subjects necessary for scientific validity.

A second study that, given the structure, allows us to predict the biological activity is through analysis of structural similarity by SwissTargetPrediction and PassOnline, when analyzing the set of five candidates proposed for repositioning, they trigger the BACE-1 by structural similarity, and effects on the central nervous system by PassOnline, which increases the probability of the existence of these candidates (see Appendix A). They also present other interaction cards associated with the central nervous system associated with Alzheimer’s, such as the muscarinic receptor M2, M5, serotonin and dopamine receptors, and serotonin and dopamine transporter, which leads to other application approaches for these candidates and not a classic mono target route but with a multitarget approach. The search for new treatment alternatives against Alzheimer’s is of great relevance worldwide, it stands out for being multifactorial, attacking various pathways, the results obtained suggest that BACE-1 is a highly relevant target given that it has the potential to reduce the progression of the disease; likewise, it can be combined with other associated proteins, such as γ-secretase, acetylcholinesterase, MAO-A, among others. Although the significant role of BACE-1 inhibitors in the treatment of neurodegenerative diseases, BACE-1 has been associated with other pathologies, so the candidates for repositioning have a multi-target nature and allow advance for new studies.

## 3. Materials and Methods

### 3.1. Substrate, Inhibitors and DrugBank Data Base Preparation

All drugs from the DrugBank database [38], the reference inhibitor for BACE-1 were optimized in Macromodel [40] and brought to physiological conditions of pH 7.4 in LigPrep [41] with RMSD of 0.3 A according to the previously reported protocol [42].

### 3.2. Protein Preparation

For BACE-1 (6C2I, [23]) it was cleaned up and prepared in Maestro and brought to pH physiological conditions in Protein Preparation Wizard [43] according to the previous protocol [42].

### 3.3. Molecular Docking

Molecular docking was performed in the Glide module [44] validating by re-docking with the cocrystal, obtaining RMSD < 1.0 A. Database docking was performed at three levels, HTVS, SP, and XP; for flexibility in hydroxyl groups by Ser, Thr, and Try, and the thiol in Cys residues, according to the previously reported protocol [42].

### 3.4. ADME and Toxicological Properties and 2D Similitud Structure Prediction

Data was obtained from DrugBank database [38] and predicted by QikProp [45] from Schrodinger. Toxicological data obtained in DrugBank and bibliography. For the prediction of activity by structural similarity, Swiss target prediction [46] and Pass Online [47] from Way2Drug were used to determine activities associated with the central nervous system.

## 4. Conclusions

BACE-1 is an ideal target for the search for new treatments against Alzheimer’s. Although Verubecestat and EJ7 are good candidates, the first has unwanted adverse and tolerance effects and the second is in preclinical studies. Therefore, bioinformatics-assisted drug repositioning proved to be a viable alternative, analyzing the coupling energy, ADME, and toxicological properties, five candidates were found, gluphenazine, naratripan, bazedoxifene, frovatriptan, and raloxifene, denoting the indole or thioindole group as possible pharmacophore and specifically hydrogen bond interactions with Asp31 and Asp 227, and aromatic with Try70 into the catalytic site of BACE-1. The structure of these drugs must be amphipathic, the hydrophobic part being preferably aromatic, these as structural requirements obtained in the present study. Finally, the design of new molecules such as BACE-1 inhibitors and the preclinical evaluation of the proposed candidates is highly recommended.

## Figures and Tables

**Figure 1 ijms-24-08164-f001:**
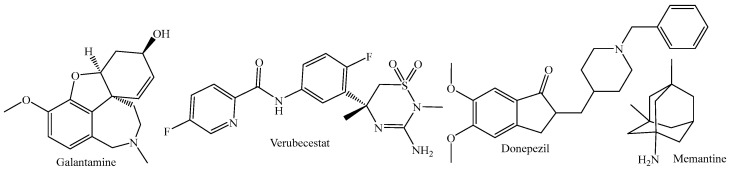
Drugs associated with molecular targets related to Alzheimer’s.

**Figure 2 ijms-24-08164-f002:**
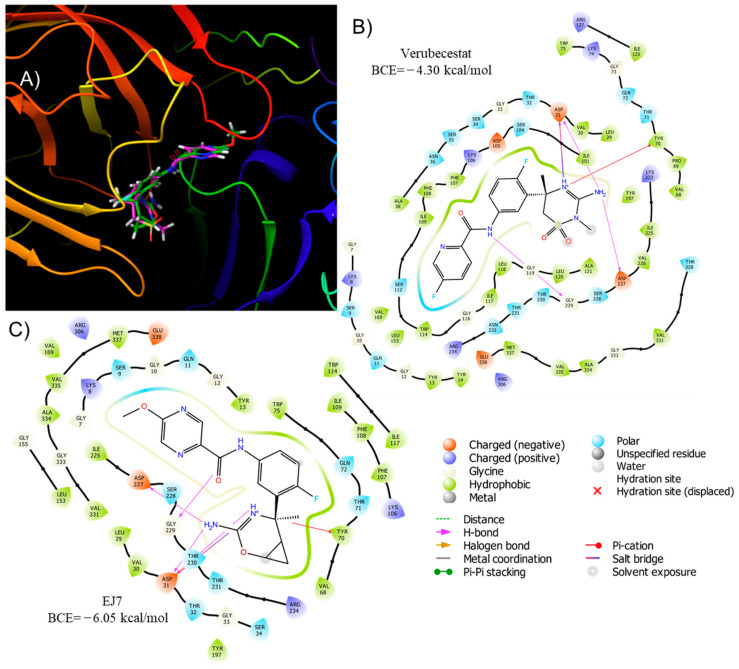
(**A**) 3D overlay of Verubecestat (green) and EJ7 (pink) at the BACE-1 catalytic site. 2D interaction at the catalytic site of BACE-1 (**B**) Verubecestat, (**C**) EJ7.

**Figure 3 ijms-24-08164-f003:**
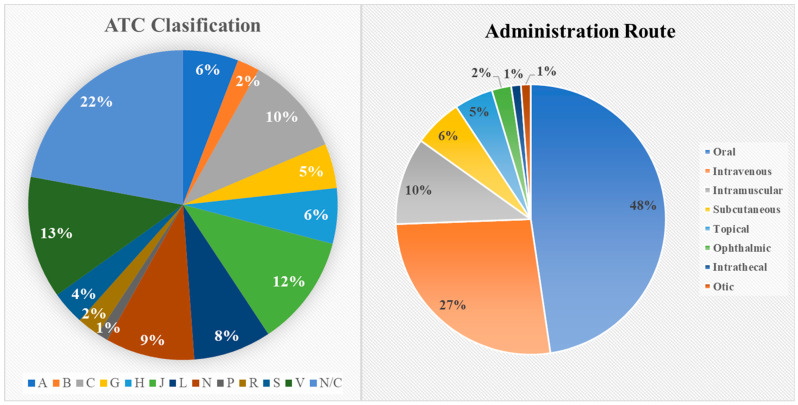
ATC classification and routes of administration of drugs with better BCE than EJ7.

**Figure 4 ijms-24-08164-f004:**
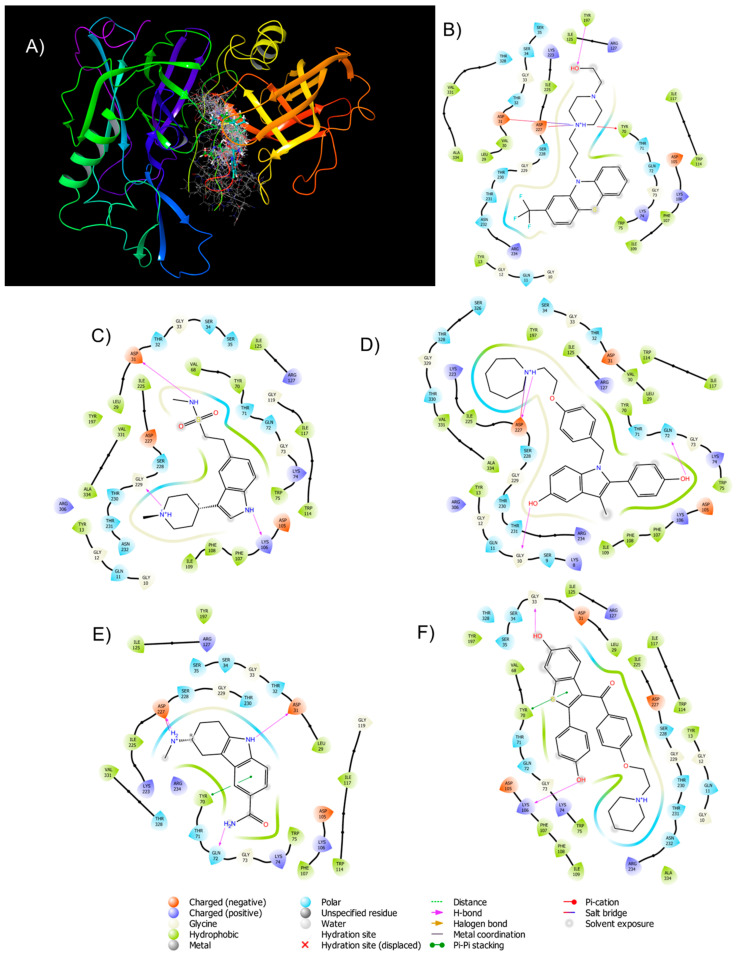
BACE-1 interaction with (**A**) 5 reposition candidates and reference inhibitors in 3D, 2D interaction at the catalytic site of BACE—(**B**) Fluphenazine, (**C**) Naratripan, (**D**) Bazedoxifene, (**E**) Frovatriptan, and (**F**) Raloxifene.

**Table 1 ijms-24-08164-t001:** Binding coupling energy and traditional uses of Drugbank database vs. BACE-1 inhibitors with oral administration.

DrugBank ID	Name	Binding Energy (kcal/mol)	Clinical Uses
DB14642	Lypressin	−8.669	Diabetes insipidus
DB00035	Desmopressin	−8.298	Antidiuretic, antihemophilic, and von Willebrand disease
DB06636	Isavuconazonium	−7.587	antifungal
DB01232	Saquinavir	−7.501	HIV-1
DB00104	Octreotide	−7.156	Acromegaly and diarrhea-associated carcinoid tumors
DB11190	Pantethine	−6.965	dietary supplement
DB11691	Naldemedine	−6.946	Opioid-induced constipation
DB14753	Hydroxystilbamidine	−6.689	Blastomycosis, antifungal, and antitrypanosomal
DB00952	Naratriptan	−6.648	Migraine
DB00998	Frovatriptan	−6.612	Migraine
DB04703	Hesperidin	−6.602	Hemorrhoids, mild allergies
DB00430	Cefpiramide	−6.595	Antibiotic
DB01288	Fenoterol	−6.550	Asthma
DB00295	Morphine	−6.467	acute and chronic pain
DB01608	Periciazine	−6.362	Treatment of aggressiveness, impulsivity, and hostility
DB00481	Raloxifene	−6.36	Prevention of osteoporosis and breast cancer
DB08907	Canagliflozin	−6.335	Diabetes mellitus type 2
DB00350	Minoxidil	−6.334	Hypertension
DB00284	Acarbose	−6.274	Diabetes mellitus type 2
DB00489	Sotalol	−6.249	ventricular arrhythmias
DB00623	Fluphenazine	−6.24	Neuroleptic
DB00221	Isoetharine	−6.204	Emphysema, bronchitis, and asthma
DB00118	Ademetionine	−6.18	Chronic liver disease, depression, and osteoarthritis
DB00370	Mirtazapine	−6.168	antidepressant
DB12141	Gilteritinib	−6.14	relapsed acute myeloid leukemia
DB08995	Diosmin	−6.135	vascular enhancement
DB00850	Perphenazine	−6.119	Schizophrenia
DB12783	Benserazide	−6.118	Parkinson
DB12015	Alpelisib	−6.09	Cancer
DB01627	Lincomycin	−6.075	Antibiotic
DB00179	Masoprocol	−6.073	actinic keratoses
DB06401	Bazedoxifene	−6.07	menopause and osteoporosis
DB04861	Nebivolol	−6.061	Hypertension and heart failure
DB12285	Verubecestat	−4.298	Alzheimer’s disease: *β*-secretase-1 inhibitor
Not Assigned	EJ7	−6.050	Alzheimer’s Disease Clinical trials

**Table 2 ijms-24-08164-t002:** ADME properties of reposition candidates to BACE-1 inhibitors.

Name	Mol Weight	LogP	LogS	H-Bond (D/A)	Lipinsky Rule	QPlogBB *	CNS *	HOA *	#metab	PSA *	Jm *
Desmopressin	1069.22	−5.81	−10.59	12.5/26.3	3	−10.59	−2	1	18	493.82	0.000554
Isavuconazonium	717.77	1.73	−5.16	2.0/9.0	3	−0.61	0	1	15	159.37	0.000124
Saquinavir	670.85	2.56	−3.61	5.0/13.7	3	−2.13	−2	2	7	181.08	0.006794
Octreotide	1019.24	−0.18	−0.05	9.5/20.1	3	−6.01	−2	1	17	351.65	0.006562
Pantethine	554.72	−1.30	−1.09	6.0/15.8	3	−4.48	−2	1	6	221.48	0.398822
Naldemedine	570.64	2.67	−3.86	2.0/10.8	1	−1.23	−2	2	6	146.65	0.000286
Hydroxystilbamidine	280.33	0.80	−2.23	7.0/3.8	1	−2.62	−2	2	1	117.38	0.000005
Naratriptan	335.46	2.19	−3.69	2.0/6.5	0	−0.56	1	3	3	70.45	0.000582
Frovatriptan	243.31	0.88	−1.97	4.0/4.0	0	−0.48	0	3	4	79.59	0.005498
Hesperidin	610.57	−1.25	−3.31	7.0/20.1	3	−4.20	−2	1	11	237.30	0.00026
Cefpiramide	612.63	1.70	−5.68	2.3/13.0	2	−4.72	−2	1	7	260.97	0.0
Fenoterol	303.36	0.77	−1.36	5.0/5.5	0	−1.69	−2	2	7	102.59	0.015216
Raloxifene	473.59	4.14	−4.35	2.0/6.3	0	−0.72	0	3	5	77.06	0.000973
Fluphenazine	437.52	4.33	−4.13	1.0/6.2	0	0.66	2	3	7	36.16	0.000174
Isoetharine	239.31	0.52	−1.17	4.0/4.7	0	−0.71	−1	2	4	74.28	0.368272
Gilteritinib	552.72	3.16	−4.86	3.0/12.0	2	−0.55	0	2	6	117.52	0.000001
Diosmin	608.55	−1.22	−3.23	7.0/19.8	3	−4.23	−2	1	9	237.23	0.000273
Benserazide	257.25	−1.71	0.10	7.0/7.5	1	−2.11	−2	2	8	154.48	0.015535
Alpelisib	441.47	2.37	−4.60	3.0/7.0	0	−0.81	−1	3	6	108.59	0.003243
Lincomycin	406.54	0.19	−1.33	5.0/13.5	0	−1.30	−2	2	7	120.41	0.052177
Masoprocol	302.37	2.30	−2.94	4.0/3.0	0	−1.69	−2	3	6	88.39	0.060709
Bazedoxifene	470.61	5.86	−6.68	2.0/4.3	1	−0.69	1	1	6	57.80	0.000017

* QPlogBB, predicted brain/blood partition coefficient, CNS, predicted central nervous system activity, HOA, Human oral adsorption, #metab, metabolites number, PSA, Van der Waals surface area of polar atoms, Jm, Predicted maximum transdermal transport rate.

**Table 3 ijms-24-08164-t003:** Amino acid residue interaction by repositioning of drug candidates to *β*-secretase-1.

Name	Charged (+)	Charged (−)	Gly	Polar	Hydrophobic	H-Bond	Pi-Pi Stacking	Pi-Cation	Salt Bidge
Fluphenazine	Lys74, Lys106, Arg127, Lys223, Arg234	Asp31, Asp105, Asp227	Gly10, Gly12, Gly33, Gly73, Gly229	Gln11, Thr32, Ser34, Ser35, Thr71, Gln72, Ser228, Thr230, Thr231, Asn232, Thr328	Try13, Leu29, Val30, Try70, Trp75, Phe107, Ile109, Trp114, Ile117, Try197, Ile225, Val331, Ala334	Try197	-	Try70	Asp31, Asp227
Naratriptan	Lys74, Lys106, Arg127, Arg306	Asp31, Asp105, Asp227	Gly10, Gly12, Gly33, Gly72, Gly119, Gly229	Gln11, Thr32, Ser34, Ser35, Thr71, Gln72, Ser228, Thr230, Thr231, Asn232	Leu29, Val68, Try70, Try197, Ile225	Asp31, Lys106, Gly229	-	-	-
Frovatriptan	Lys74, Lys106, Arg127, Lys223, Arg234	Asp31, Asp105, Asp227	Gly33, Gly119, Glu229	Thr32, Ser34, Ser35, Thr71, Gln72, Ser228, Thr230, Thr328	Leu29, Trp75, Phe107, Trp114, Ile117, Ile125, Try197, Ile225, Val331	Asp31, Asp227	Try70	Asp227	-
Raloxifene	Lys74, Lys106, Arg127, Arg234	Asp31, Asp105, Asp227	Gly10, Gly12, Gly229	Gln11, Ser34, Ser35, Thr71, Gln72, Ser228, Thr230, Thr231, Asn232, Thr328	Try13, Leu29, Val68, Trp75, Phe107, Phe108, Ile109, Trp114, Ile117, Ile125, Try197, Ile225, Ala334	Gly33, Lys106	Try70	-	-
Bazedoxifene	Lys8, Lys74, Lys106, Arg127, Lys223, Arg234, Arg306	Asp31, Asp105, Asp227	Gly10, Gly12, Gly33, Gly73	Ser9, Gln11, Thr32, Ser34, Thr71, Gln72, Thr230, Thr231, Ser326, Thr330, Thr328	Try13, Leu29, Val30, Try70, Trp75, Phe107, Phe108, Ile109, Trp114, Ile117, Ile127, Try197, Ile225, Val331, Ala334	Gly10, Gln72, Asp227	-	-	Asp227
Verubecestat	Lys8, Lys74, Lys106, Arg127, Lys223, Arg234	Asp31, Asp105, Asp227, Glu338	Gly10, Glu12, Gly33, Gly76, Gly119, Gly229, Glu333	Ser9, Gln11, Thr32, Ser34, Ser35, Asn36, Thr71, Gln72, Ser112, Ser228, Thr230, Thr231, Asn232, Thr328	Try13, Leu29, Val30, Ala38, Val68, Pro69, Try70, Trp75, Phe107, Phe108, Ile109, Trp114, Ile117, Ile125, Val331, Ala334, Val335	Asp31(2), Asp227, Gly229	-	Try70	Asp31
EJ7	Lys8, Lys74, Lys106, Arg127, Arg234, Arg306	Asp31, Asp227, Glu338	Gly7, Gly10, Gly12, Gly33, Gly73, Gly115, Gly119, Gly229, Gly333	Ser9, Gln11, Thr32, Ser34, Ser35, Asn36, Thr71, Gln72, Ser112, Ser228, Thr230, Thr231, Asn232	Leu29, Val30, Ala38, Val68, Pro69, Try70, Trp75, Phe107, Phe108, Ile109, Trp114, Ile117, Leu153, Ile225, Val331, Ala334, Val335, Met337	Asp31(2), Asp227, Gly229	-	Try70	Asp31

Red, present in Verubecestat and EJ7, Blue, present only Verubecestat, and Green, present only EJ7.

**Table 4 ijms-24-08164-t004:** Adverse effects reported by *β*-secretase-1 reposition drug candidates.

Name	Adverse Effects
Fluphenazine	Drowsiness, lethargy, dizziness, lightheadedness, nausea, loss of appetite, sweating, dry mouth, blurred vision, headache, constipation.
Naratriptan	Severe chest pain, heaviness, tightness or pain in the chest, throat, and sleepiness, burning sensation, tingling, numbness, seizures.
Frovatriptan	Sudden severe stomach pain, bloody diarrhea, severe chest pain, difficulty breathing, irregular heartbeat, cramps, numbness, chest pain or pressure, agitation, hallucination, hyperactive reflexes, diarrhea.
Raloxifene	Hot flashes, leg cramps, swelling in the hands, feet, ankles, or lower legs, generates flu syndrome, joint pain, sweating, difficulty sleeping.
Bazedoxifene	Nausea, heartburn, stomach pain, diarrhea, muscle spasms, neck, and throat pain
Verubecestat	Injuries, falls, suicidal ideation, weight loss, sleep disorders, rashes, hair color change, dizziness, anxiety.
EJ7	No reports by preclinical studies.

## Data Availability

The data that support the findings of this study are available upon request.

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
