# Peer review of "β-Secretase-1: In Silico Drug Reposition for Alzheimer’s Disease"

_ijms, 2023, doi:10.3390/ijms24098164_

Round 1

Reviewer 1 Report

Increased β-secretase (BACE1) activity has been associated with neurodegeneration and accumulation of amyloid precursor protein (APP) products in Alzheimer’s disease (AD). Thus, the inactivation of BACE1 could be important in the treatment of AD. And drug repositioning and repurposing can enhance traditional drug development efforts and could accelerate the identification of new treatments for individuals with AD dementia and mild cognitive impairment.

This study (ijms-2347264) aims to identify new potential candidates for the treatment of Alzheimer's, with the help of in silico studies such as molecular docking and ADME prediction, from a broad list of candidates provided by the Drug Bank database through bioinformatics-assisted drug repositioning. Of 1145 drugs capable of interacting with the enzyme with higher coupling energy, five candidate drugs ‒ fluphenazine, naratriptan, bazedoxifene, frovatriptan, and raloxifene which exhibit pharmacophore-specific features, including the indole or thioindole group, and interactions with key amino acids, were identified as possible BACE-1 inhibitors. Overall, this study provides insights into the potential use of in silico methods for drug repurposing and identification of new candidates for the treatment of Alzheimer's disease (targeting BACE1) and proposes a preclinical evaluation of the proposed drug candidates.

The study is interesting and needs to go minor revisions for publication by IJMS.

1.         Why is QUD (a non-peptic catalytic BACE1 ligand) and PMF (a noncompetitive inhibitor of BACE1) used as reference drug for comparison of docking data?

2.         Using in silico methods for drug repurposing, five candidate drugs were identified as new candidates for the treatment of Alzheimer's disease from a pool of 1145 drugs. Did you ever check the BACE1 inhibition activity of those five candidates? Are they potent enough to be considered for preclinical evaluation? What about their inhibition mechanism or kinetics?

3.         Quality of Figures 2 and 4 is relatively low and it's quite difficult to distinguish amino acid residues. I would suggest enlarging the letters/abbreviations denoting amino acids.

4.         English edition-suggested.

English edition-suggested.

Author Response

Reviewer 1

  1. Why is QUD (a non-peptic catalytic BACE1 ligand) and PMF (a noncompetitive inhibitor of BACE1) used as reference drug for comparison of docking data?

Response: These two were selected given the extremes that represent the Verubecestat which presents the lowest energy with reported biological effect in addition to clinical studies and the EJ7 that has the superior effect compared with inhibition studies; the results obtained from the complete BACE1 study with the other inhibitors for consultation and better establishment of the key amino acids.are also added at the manuscript as supplementary material.

  1. Using in silico methods for drug repurposing, five candidate drugs were identified as new candidates for the treatment of Alzheimer's disease from a pool of 1145 drugs. Did you ever check the BACE1 inhibition activity of those five candidates? Are they potent enough to be considered for preclinical evaluation? What about their inhibition mechanism or kinetics?

Response: The coupling energy is directly related to the strength as an inhibitor, which suggests that the proposed candidates could be excellent for in vitro and/or in vivo studies , accompanied by this, it will be possible to study the mechanism of BACE-1 inhibition and kinetics, although this study proposes that they would function as BACE-1 competitive inhibitors. In the supplementary material, the inhibition data for the controls are added to support the correlation between the BCE and inhibition values.

  1. Quality of Figures 2 and 4 is relatively low and it's quite difficult to distinguish amino acid residues. I would suggest enlarging the letters/abbreviations denoting amino acids.

Response: Thanks for your recommendation. The quality was improved to 600 dpi to allow a higher resolution zoom and analyze the interaction amino acids.

  1. English edition-suggested.

Response: Thanks for your advice. The entire document was revised, correcting several spelling and writing errors.

Reviewer 2 Report

Authors reported in silico analysis for the identification of potential active compounds against BACE-1. Methods are well described, and overall, the analyses performed are clear and detailed.

On the other hand, I would not consider a repurposing study with just molecular docking analysis as sound and robust for the scientific community. In this respect, I suggest to add further analysis in order to obtain a "consensus" strategy with different computational approaches, at least for the best resulted compounds: for istance, there are free web platform such as PLATO (https://doi.org/10.3390/ijms23095245, https://doi.org/10.1021/acs.jcim.1c00498) that was designed for repurposing goals and uses ligand-based approach. It could be interesting if the same compounds resulted from analyses the authors have already performed were confirmed as potential BACE-1 binders. 

Furthermore, other known BACE-1 inhibitors can be used as references for defining and explaining key interactions returned from molecular docking analysis.

Author Response

Reviewer 2

  1. On the other hand, I would not consider a repurposing study with just molecular docking analysis as sound and robust for the scientific community. In this respect, I suggest to add further analysis in order to obtain a "consensus" strategy with different computational approaches, at least for the best resulted compounds: for istance, there are free web platform such as PLATO (https://doi.org/10.3390/ijms23095245, https://doi.org/10.1021/acs.jcim.1c00498) that was designed for repurposing goals and uses ligand-based approach. It could be interesting if the same compounds resulted from analyses the authors have already performed were confirmed as potential BACE-1 binders. 

Response: We appreciate your recommendation and have revised the results from our manuscript as suggested. The results obtained by SwissTargetPrediction were added, where BACE-1 comes to light due to structural similarity, as well as other proteins associated with CNS and the development and treatment against Alzheimer's, thus increasing the reliability of the theoretical results through two approaches. In addition PASS ONLINE results from Way2Drug were added given the correlation with interaction in CNS as possible activities, an attempt was made to add the results of PLATO, but unfortunately the server seems to be undergoing maintenance since it did not allow access to it.

  1. Furthermore, other known BACE-1 inhibitors can be used as references for defining and explaining key interactions returned from molecular docking analysis.

Response: Thanks for your suggestion, so we decide that the results obtained from the other inhibitors are also added to the supplementary material for consultation and better establishment of the key amino acids.